# Early Diagnostic Markers of Late-Onset Neonatal Sepsis

Preslava Gatseva [1,2,*], Alexander Blazhev [3], Zarko Yordanov [4] and Victoria Atanasova [1,2]

[1] Department of Pediatrics, Medical University Pleven, 5800 Pleven, Bulgaria; umbal@umbalpln.com
[2] Dr. Georgi Stranski University Hospital, 5800 Pleven, Bulgaria
[3] Department of Anatomy, Histology, Cytology and Biology, Medical University Pleven, 5800 Pleven, Bulgaria; alexander.blazhev@mu-pleven.bg
[4] Department of Anaesthesiology and Resuscitation, Medical University Pleven, 5800 Pleven, Bulgaria; zyordanoff@yahoo.com
* Correspondence: pediatricclinicpleven@gmail.com or preslava_gatseva@abv.bg; Tel.: +359-878838439

**Abstract:** Objective: Early diagnosis of nosocomial infections in newborns is a great challenge, because in the initial phase of systemic infection, clinical symptoms are often non-specific, and routinely used hematological markers are not sufficiently informative. The aim of this study was to determine the potential of early inflammatory markers to diagnose late-onset neonatal sepsis—procalcitonin (PCT), interleukin 6 (IL-6), interleukin 8 (IL-8) and endocan (ESM-1). Material and methods: A prospective clinical–epidemiological study was conducted in a third-level NICU in Pleven, Bulgaria. Patients with suspected late-onset sepsis and healthy controls were tested. A sandwich ELISA method was used to measure the serum concentrations of biomarkers. Results: Sixty newborns were included, of which 35% symptomatic and infected, 33.3% symptomatic but uninfected and 31.7% asymptomatic controls. The mean values of PCT, IL-6, I/T index and PLT differ significantly in the three groups. For ESM-1, IL-8 and CRP, the difference was statistically insignificant. The best sensitivity (78%) and negative predictive value (84%) was found for IL-6. The combinations of PCT + IL-6 and PCT + IL-6+ I/T+ PLT showed very good diagnostic potential. Conclusion: The introduction into the routine practice of indicators such as PCT and IL-6 may provide an opportunity to promptly optimize the diagnostic and therapeutic approach to LOS.

**Keywords:** neonatal sepsis; early diagnosis; inflammatory markers; procalcitonin; interleukins; endocan

## 1. Introduction

Nosocomial infections are a serious problem in neonatal intensive care units—they are seen as one of the main causes of morbidity and mortality among neonates. They develop after the 72nd hour of admission and occur in relation to the patient's medical care [1]. A major concern among these infections is the late neonatal sepsis, which refers to an infection involving the bloodstream and leading to abnormalities in the systemic circulation [2]. Since the early 1980s, epidemiological studies have observed a general increase in the incidence of late-onset sepsis (LOS) in newborns together with an improvement in the survival of preterm infants, especially those with very low birth weight (VLBW), which is also associated with the use of various medical devices and invasive procedures in the neonatal intensive care unit (NICU) [3].

The degree of the clinical manifestation of LOS can be highly variable, depending on the virulence of pathogens and the host defense mechanisms. The initial symptoms such as temperature instability, apnea, tachypnea, tachycardia, dyspnea, hyper- and hypothermia, feeding difficulties and irritability, are generally non-specific and difficult to differentiate from other non-infectious conditions [4]. They can be predictors of late-onset neonatal sepsis (LOS) and, if not recognized on time, can lead to severe complications such as shock and death. Additionally, in the long term, it can be the cause of growth retardation, neurodevelopmental impairment and bronchopulmonary dysplasia [5].

To begin with, despite the fact that several clinical and hematological markers are usually considered to diagnose systemic infection, the correct combination is still not clearly defined. In routine practice, total white blood cell count, elevated immature-to-total polymorphonuclear cell ratio, platelet count and acute-phase protein (C-reactive protein) are tested, and the gold standard is a positive microbiological culture of biological samples. However, these markers are not sufficiently predictive in the initial phase of systemic neonatal infection and are defined as "late" indicators. Therefore, there is a delay in the diagnosis of affected neonates, which may lead to the late initiation of treatment and the abovementioned complications. If antibiotics are started upon the suspicion of sepsis without any evidence, the list of disadvantages may include prolonged and unnecessary therapy, the selection of resistant microorganisms and increased hospital costs [6].

Novel laboratory parameters would be of paramount importance if they can trustingly guide the medical professionals in their tough decisions. The ideal biomarker should have high sensitivity to ensure that cases are not missed, with high specificity to avoid exposing unaffected infants to unnecessary treatment. Moreover, positive predictive value and negative predictive value (NPV) should also be correspondingly high. Even with several promising potential biomarkers, there is no definitive biomarker, group of biomarkers or scoring system that can be utilized exclusively at this time. Thus, novel biomarkers need individual and prospective assessment [7].

Many biomarkers for sepsis have been investigated so as to determine their diagnostic usefulness. They are basically grouped as acute-phase proteins, cell surface antigens, cytokines and chemokines as well as soluble adhesion molecules.

The aim of this study was to determine the potential of procalcitonin (PCT), interleukin 6 (IL-6), interleukin 8 (IL-8) and endocan (ESM-1) to diagnose late-onset sepsis in neonates. Their serum concentrations were assessed and the validation criteria for diagnostic tests—sensitivity, specificity, accuracy, positive and negative predictive value—were analyzed. Their diagnostic value was compared to that of conventional markers like C-reactive protein (CRP), immature-to-total polymorphonuclear cell ratio (I:T index), platelet count (PLT) and white blood cells count (WBC). Combinations of these tests were also considered.

## 2. Materials and Methods

A prospective clinical–epidemiological study (January 2022–January 2023) was conducted in the Clinic of Neonatology of the University Hospital "Dr. Georgi Stransky", Pleven. The study included preterm and term neonates, with a stay in the neonatal intensive care unit (NICU), which exceeded 72 h. Exclusion criteria were the presence of severe congenital anomalies and an early postoperative period. The work group consisted of patients with new onset symptoms, which led to the suspicion of LOS. As per the inclusion criteria, we accepted: (1) the presence of at least three clinical and laboratory indicators and (2) at least one risk factor, suggestive of acquired NICU neonatal infection (Table 1).

**Table 1.** Clinico-laboratory indicators and risk factors, suggestive of acquired NICU neonatal infection.

| Clinical Signs (New-Onset): | Laboratory Signs: | Risk Factors: |
|---|---|---|
| Rhythmic breathing disorders (apnea, dyspnea) | Low/high WBC count | Mechanical ventilation |
| Increased oxygen needs | Thrombocytopenia | Central venous line |
| Requirement of respiratory support | Positive prophylactic microbiological testing | Parenteral nutrition |
| Skin color changes (pale/greyish skin) | Hyper/hypoglycemia | Urinary catheter |
| Tachy/bradycardia | Metabolic acidosis | Gastral tube |

**Table 1.** *Cont.*

| Clinical Signs (New-Onset): | Laboratory Signs: | Risk Factors: |
|---|---|---|
| Abdominal distension | | Postnatal corticosteroids |
| Vomiting | | Chronic respiratory and heart failure |
| Diarrhea | | |
| Decreased motor activity | | |
| Depressed consciousness | | |
| Irritability, seizures | | |
| Bulged fontanelle | | |
| Jaundice | | |
| Weight loss | | |
| Hypothermia/hyperthermia | | |

The case group was then divided into two subgroups:

- Group 1—symptomatic and infected patients (patients with proven LOS). Diagnoses of clinical sepsis (with negative microbiological results) and septicemia (with positive microbiological results) were accepted. The evaluation was complex, considering the follow-up hematological investigations in the next 5 days, such as elevated CRP, leukocytosis/leukopenia; results of microbiological samples; evolution of the clinical status, treatment and outcome.
- Group 2—symptomatic but uninfected patients (with no evidence of sepsis during the next 5 days of stay—control paraclinical tests were within reference limits and microbiological samples were negative). Other non-infectious condition or disease could be proven.

In order to analyze the reference ranges of the indicators, a control Group 0 of asymptomatic newborns in stable general condition with hospital stay >72 h was formed. The taken samples were part of regular follow-up blood tests.

On the day of suspicion of infection, 1.5–2 mL of blood was collected from a peripheral vein, the separated serum was stored at $-80\ ^\circ$C until analysis.

Concentrations of IL-6, IL-8, PCT and ESM-1 were measured in the laboratory of the Medical University of Pleven. Commercial kits based on sandwich ELISA (enzyme-linked immunosorbent assay) were used, respectively: Human IL-6 ELISA Kit, Invitrogen Thermo Fisher Scientific Inc; Human IL-8 ELISA Kit, Invitrogen Thermo Fisher Scientific Inc; Human Procalcitonin ELISA, BioVendor LM; Human Endothelial Cell-Specific Molecule 1 ELISA Kit, CUSABIO TECHNOLOGY LLC. The reaction was performed according to the manufacturer's instructions.

Data were entered and processed with the statistical packages IBM SPSS Statistics 25.0. and Office 2021 Excel. A significance level rejecting the null hypothesis was accepted as $p < 0.05$. The following statistical methods were applied: descriptive and graphical analyses, non-parametric Shapiro–Wilk test, Stewart's T-criterion, non-parametric Mann–Whitney test, ROC curve analysis and multiple binary logistic regression analysis

## 3. Results

A total of sixty newborns were enrolled: 42 (70%) boys and 18 (30%) girls. The mean gestational age (g.a.) of the study population was $29.75 \pm 3.61$ gestational weeks (g.w.), ranging between 25 and 40 g.w.

The distribution of study population by groups is presented in Figure 1.

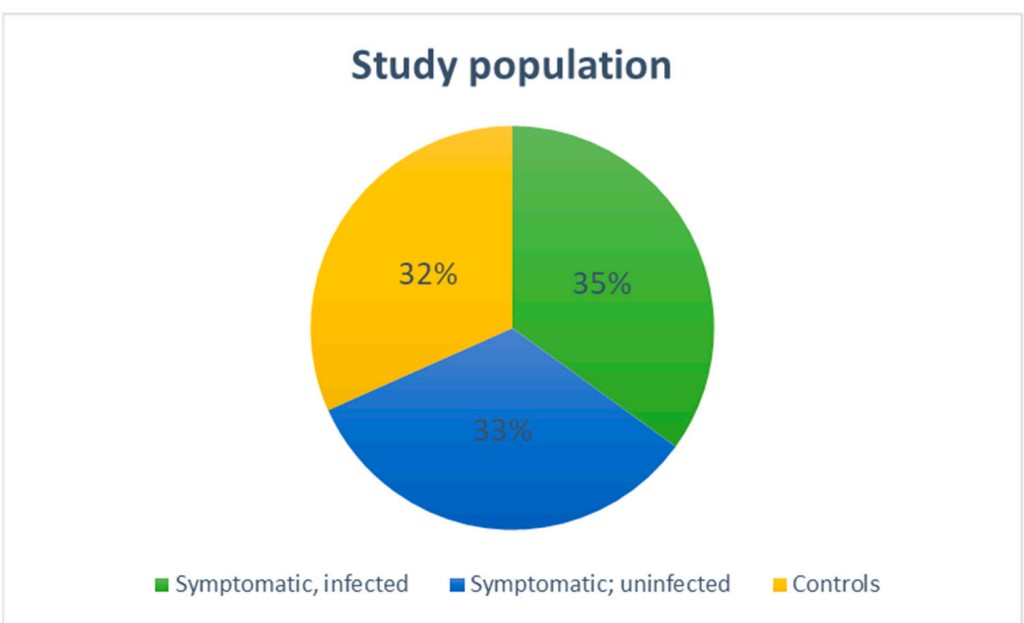

**Figure 1.** Distribution of the study population by groups.

Microbiological samples were positive only in 66% of the infected patients (14/21). The most common pathogen, responsible for LOS was Gram-negative *Klebsiella pneumoniae* (7/14, 50%), followed by Gram-positive *Staphylococci* sp. (3/14, 21%). Less common pathogens were *Escherichia coli*, *Serratia* sp. and *Enterobacter* sp.

The comparative analysis of the groups, according to the values of markers PCT, ESM-1, IL-6 and IL-8, and the indicators CRP, I/T index, WBC and PLT, revealed that the three groups differed significantly in four of the indicators included in the table, namely PCT, IL-6, I/T index and PLT (Table 2).

In septic neonates compared to non-septic from groups 0 and 2, the mean serum concentration (ng/mL) of procalcitonin was significantly higher ($2.27 \pm 3.22$ vs. $0.43 \pm 0.57$, $p = 0.004$ and $0.44 \pm 0.58$, $p = 0.005$, respectively). A similar correlation was found between the mean IL-6 serum concentration ($30.72 \pm 53.48$) and the other two groups, whose arithmetic means were not different from each other ($29.82 \pm 104.02$ and $5.07 \pm 5.39$) (Table 2).

Group 1 had a significantly higher mean I:T index than Group 2 ($0.35 \pm 0.16$ vs. $0.24 \pm 0.11$, $p = 0.024$), but not higher than controls ($0.29 \pm 0.21$), whose means were not statistically different from those of the other two groups.

The difference between the main groups for the markers ESM-1, IL-8, CRP and WBC groups was statistically negligible ($p > 0.005$).

The study groups differed significantly in the platelet count. In Group 1, they had a significantly lower mean value than that of the controls ($265.57 \pm 167.86$ vs. $384 \pm 136.13$, $p = 0.013$), but not than that of the symptomatic uninfected patients ($321.90 \pm 126$), which was not statistically different from those of the other two groups.

At the same time, we analyzed the pathological values of WBC and PLT. Table 3 shows that Group 1 is significantly different from the other two groups in both parameters, which have statistically higher relative proportions in this group. The difference between the percentages of pathological values of platelets and leukocytes in Group 0 and Group 2 is insignificant.

**Table 2.** Comparative analysis of the main groups according to the values of the markers PCT (ng/mL), ESM-1 (pg/mL), IL-6 (pg/mL), IL-8 (pg/mL) and the indices CRP (mg/L), I/T index, PLT ($\times 10^9$/L) and WBC ($\times 10^9$/L).

| Parameter | Group | N | $\bar{X}$ | SD | *p* 0–1 | *p* 0–2 | *p* 1–2 |
|---|---|---|---|---|---|---|---|
| PCT | 0. Control | 17 | 0.43 [a] | 0.57 | **0.004** | 0.858 | **0.005** |
| | 1. Symptomatic infected | 21 | 2.27 [b] | 3.22 | | | |
| | 2. Symptomatic uninfected | 18 | 0.44 [a] | 0.58 | | | |
| ESM-1 | 0. Control | 19 | 118.21 [a] | 84.00 | 0.130 | 0.967 | 0.092 |
| | 1. Symptomatic infected | 21 | 163.52 [a] | 125.11 | | | |
| | 2. Symptomatic uninfected | 20 | 116.00 [a] | 85.33 | | | |
| IL-6 | 0. Control | 17 | 29.81 [a] | 104.02 | **0.015** | 0.940 | **0.005** |
| | 1. Symptomatic infected | 18 | 30.72 [b] | 53.48 | | | |
| | 2. Symptomatic uninfected | 20 | 5.07 [a] | 5.39 | | | |
| IL-8 | 0. Control | 13 | 45.25 [a] | 106.80 | 1.000 | 0.894 | 0.705 |
| | 1. Symptomatic infected | 14 | 141.98 [a] | 346.59 | | | |
| | 2. Symptomatic uninfected | 12 | 27.62 [a] | 40.20 | | | |
| CRP | 0. Control | 18 | 3.96 [a] | 2.78 | 0.055 | 0.186 | 0.183 |
| | 1. Symptomatic infected | 20 | 21.24 [a] | 30.13 | | | |
| | 2. Symptomatic uninfected | 20 | 4.83 [a] | 2.52 | | | |
| I/T index | 0. Control | 17 | 0.29 [ac] | 0.21 | 0.117 | 0.732 | **0.024** |
| | 1. Symptomatic infected | 20 | 0.35 [bc] | 0.16 | | | |
| | 2. Symptomatic uninfected | 18 | 0.24 [a] | 0.11 | | | |
| Platelets PLT ($\times 10^9$/L) | 0. Control | 19 | 384.00 [a] | 136.13 | **0.013** | 0.147 | 0.095 |
| | 1. Symptomatic infected | 21 | 265.57 [bc] | 167.86 | | | |
| | 2. Symptomatic uninfected | 20 | 321.90 [ac] | 126.00 | | | |
| White blood cells WBC ($\times 10^9$/L) | 0. Control | 19 | 12.59 [a] | 4.01 | 0.708 | 0.667 | 0.855 |
| | 1. Symptomatic infected | 21 | 16.87 [a] | 13.34 | | | |
| | 2. Symptomatic uninfected | 20 | 13.20 [a] | 4.54 | | | |

Identical letters on the verticals indicate the absence of a significant difference; different letters indicate the presence of a significant difference ($p < 0.05$).

**Table 3.** Comparative analysis of the main groups, according to the pathological values of the indicators PLT and WBC.

| Parameter | Frequency | 0. Control | 1. Symptomatic Infected | 2. Symptomatic Uninfected | *p* 0–1 | *p* 0–2 | *p* 1–2 |
|---|---|---|---|---|---|---|---|
| PLT < 150 ($\times 10^9$/L) | n | 0 | 7 | 1 | **0.006** | 0.330 | **0.022** |
| | % | 0 [a] | 33.3 [b] | 5.0 [a] | | | |
| WBC <5 and >21 ($\times 10^9$/l) | n | 1 | 10 | 2 | **0.003** | 0.587 | **0.009** |

Identical letters on the verticals indicate the absence of a significant difference; different letters indicate the presence of a significant difference ($p < 0.05$).

According to the results in Table 2, we could pool the controls with Group 2 (symptomatic uninfected) when searching for threshold values. To determine whether statistically significant threshold values existed for the markers PCT, ESM-1, IL-6, IL-8 and the indices CRP, I/T index and PLT, ROC curve analysis was applied, distinguishing Group 1 from groups 0 and 2. From Figure 2, it is clear that PCT, IL-6, I/T index and PLT have significant threshold values. Their selection was performed according to Youden index = [maximum (sensitivity + specificity-1)]. In addition, the abnormal val-

ues of PCT ($\geq$0.5) and I/T index ($\geq$0.25) were also evaluated, as was the combination of the four variables PCT $\geq$ 0.46 + IL-6 $\geq$ 4.97 + I:T index $\geq$ 0.335 + Plt $\leq$ 161.5. Multiple binary logistic regression analysis was applied to the combination. The area under the curve was AUC = 0.880, $p < 0.001$. Furthermore, the easier option for implementation in clinical practice was tested with the two variables having the highest AUC values— PCT $\geq$ 0.46 + IL-6 $\geq$ 4.97. Multiple binary logistic regression analysis was applied to the combination. The area under the curve was AUC = 0.803, $p < 0.001$

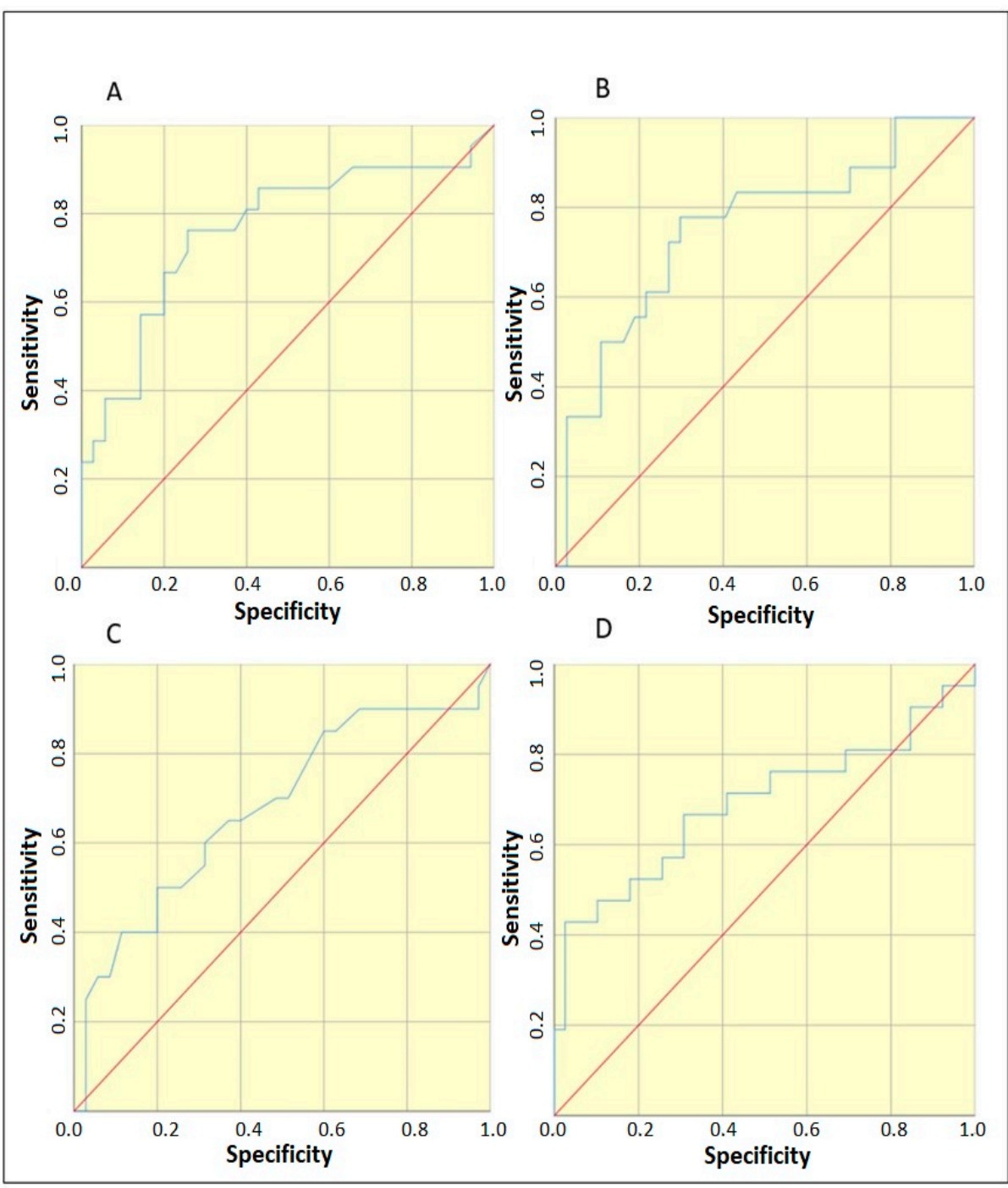

**Figure 2.** ROC curves to determine a threshold value for distinguishing Group 1 from groups 0 and 2 of: (**A**) PCT, area under the curve 0.763, $p = 0.001$; (**B**) IL-6, area under the curve 0.752, $p = 0.003$; (**C**) I:T index, area under the curve 0.677, $p = 0.03$ and (**D**) PLT, area under the curve 0.69, $p = 0.016$.

With the established threshold values, an analysis of diagnostic test validation criteria was performed. The results are presented in Table 4.

**Table 4.** Threshold values of PCT, IL-6, I/T index, PLT and the combination PCT ≥ 0.46 + IL-6 6 ≥ 4.97+ I:T index ≥ 0.335 + PLT ≤ 161.5 to distinguish Group 1 from groups 0 and 2, and validation criteria values.

| Parameter | Threshold Value | Sensitivity (%) | Specificity (%) | Positive Predictive Value (%) | Negative Predictive Value (%) | Accuracy (%) |
|---|---|---|---|---|---|---|
| PCT | ≥0.46 * | 76 | 74 | 64 | 84 | 75 |
| PCT | ≥0.5 | 71 | 74 | 63 | 81 | 73 |
| IL-6 | ≥27.5 * | 78 | 70 | 56 | 87 | 73 |
| I:T index | ≥0.335 * | 50 | 80 | 59 | 74 | 69 |
| I:T index | ≥0.25 | 20 | 97 | 80 | 68 | 69 |
| PLT | ≤161.5 | 43 | 97 | 90 | 76 | 78 |
| PCT + IL-6 + I:T index + PLT | ≥0.276 * | 94 | 68 | 63 | 95 | 78 |
| PCT + IL-6 | ≥0.206 * | 94 | 56 | 53 | 95 | 69 |

* These threshold values are calculated using data from the present study.

## 4. Discussion

Late-onset neonatal sepsis is defined as an infection occurring 72 h after birth. According to the literature, the incidence ranges from 0.6% to 14% of all neonates admitted to hospital [8]. Extremely preterm infants are at the highest risk, with a cited incidence of about 34%. Risk factors for LOS include prematurity, a prolonged exposure to invasive procedures, delayed enteral feeding, the need for surgical intervention and underlying respiratory and cardiac disease.

The three main challenges in the diagnosis of neonatal sepsis are: the myriad of clinical symptoms that can mimic sepsis; false-negative bacterial cultures in cases of so-called culture-negative sepsis; and the need for empiric treatment for a minimum of 24 to 48 h while cultures are incubated [9].

The complete blood count with differential count is the most commonly tested laboratory indicator of infection since it is a cheap and easy-to-perform method. In recourse-limited areas, this may be the only diagnostic tool for neonatal sepsis. This inspired a study in Ethiopia, showing that the mean of total white cell count and platelet count is lower in septic patients. Leucopenia, defined in the study as WBC less than or equal to 12,500/mm$^3$, has a sensitivity of 35.2% and a specificity of 92% in LOS and seems to be more valuable in early onset sepsis. The sensitivity and specificity of thrombocytopenia ($<145,000$/mm$^3$) were 54.1 and 70.4%, respectively [10]. Low sensitivity and good specificity are often found, considering this indicator as unreliable [11].

The combination of leucopenia and a high I/T ratio ($>0.2$) shows similar results, although they show increased odds of infection (highest odds ratios: 5.38 and 7.97, respectively). In a study by Hornik et al., they have high specificity and negative predictive value and low sensitivity [12]. They seem to be more useful in eliminating healthy newborns than identifying infected ones. Excellent 100% negative predictive value is reached in a study by Murphy et al. with two normal I/T ratio and sterile blood culture [9]. According to our results, patients with LOS have a remarkably higher incidence of leucopenia/leukocytosis and thrombocytopenia. The best specificity (97%), positive predictive value (90%) and accuracy (78%) show the platelet count parameter. A possible correlation may be found with the bacteriological agents causing sepsis in our NICU, where Gram-negative bacteria are the prevailing causative agents of nosocomial infections. This assumption is derived from some reports that show that there are differences in platelet response to infection with

the two major groups of microorganisms [13]. Unexpectedly, we found a very good performance of I/T index with a specificity of 97% and a positive predictive value of 80% (for a threshold value ≥ 0.25), which is generally considered a "late" marker of inflammation. The accuracy is also comparatively high and, as a whole, the decent performance of this marker may be attributed to the skillfulness of the technicians in the laboratory.

Furthermore, an acute-phase protein that is very often interpreted with complete blood count is the C-reactive protein. Usually, it is not positive in the early stage of infection and, as we suggested, it did not reach statistically significant values in our study. The peak of its level is after 24 h, meaning that the sensitivity for neonatal sepsis is the lowest in the early stage of infection. It rises in serial testing at the 24–48th hour after the onset of symptoms. The specificity and positive predictive value range from 93 to 100%, so it is referred to as a "specific" but "late" marker for neonatal infection [14].

In contemporary times, there have been considerable scientific explorations of diverse serum inflammatory biomarkers, with the primary objective of unveiling a remarkably sensitive "early" examination method to evaluate instances of inflammation and infection. These biomarkers encompass procalcitonin, interleukin 6, interleukin 8 and endocan (PCT, IL-6, IL-8, endocan/ESM-1). It is hypothesized that their concentrations exhibit augmentation within the initial hours of an inflammatory response. Demonstrating substantial sensitivity and diagnostic significance, these biomarkers hold potential practical utility. Nonetheless, the outcomes of prior conducted studies frequently lack conclusive evidence, and the establishment of a uniform methodology remains an unresolved endeavor [15].

Recently, PCT has been more widely used in the diagnosis of neonatal infections. Procalcitonin is a peptide, produced by monocytes and hepatocytes in response to systemic inflammation and, according to some studies, appears to be more sensitive than CRP in bacterial infections [16]. Nonspecific, i.e., infection-independent, the induction of PCT synthesis can occur after major surgery, multiple trauma and during the early neonatal period, which is why we excluded these newborns from our study [17]. In neonatal sepsis, its concentrations rise after 4 h from the proinflammatory effect of bacterial endotoxins, and reach their peak after 6–8 h, thus rising earlier than CRP [18]. Its half-life is 25–30 h and concentrations are not affected by gestational age. Its elevation is independent of calcitonin and is associated with neurotransmission, immunomodulation, vascular control during infection and in the systemic inflammatory response syndrome (SIRS) [19]. Procalcitonin is usually referred as a highly specific marker for the diagnosis and monitoring of bacterial infections and sepsis, also referring to the severity of the infectious process. In addition, PCT is an indicator of the therapeutic success—a decrease in its plasma levels 24 h after treatment initiation is connected to favorable therapeutic response [17]. Its diagnostic profile for systemic bacterial infections and necrotizing enterocolitis has shown to be superior to all other acute-phase proteins, with a sensitivity and specificity of 87 to 100%. Bustos et al. performed a prospective observational study of a cohort of 53 neonates with clinically suspected late-onset neonatal sepsis. Procalcitonin showed a sensitivity of 88%, a specificity of 71.4% and a negative predictive value of 87% [20].

In recent years in Bulgaria, there have been only a few neonatal centers that routinely test PCT levels. This is why there is insufficient research in this field—the studies focus on early onset sepsis more than LOS and the laboratory methods may be different. For example, the semi-quantitative immunochromatographic study of procalcitonin in neonates with generalized bacterial infection was carried out in 2001 by Georgieva et al. [21]. Eighty-seven neonates were studied—30 with early and late sepsis, eight with late sepsis and 49 controls. A sensitivity of 63% and a specificity of 100% were demonstrated for values above 2 ng/mL.

In our study, at a threshold value > 0.46 ng/mL, PCT showed a sensitivity of 76%, a specificity of 74% and a negative predictive value of 84%, which are close to those cited in the literature. This proves the necessity of PCT to be implemented in the basic septic screening in the country.

Another group of biomarkers that have been evaluated for the diagnosis of neonatal sepsis are the interleukins. Interleukin 6 is an important cytokine of the host's early immune response. A review of studies, conducted from 1990 to 2020 shows that IL-6 is the biomarker that has been studied more than any other interleukin in newborns [22]. Different threshold values have been proposed, and as this value increases, specificity increases at the expense of sensitivity. In most studies, the results are extremely promising. The mean (30.72) and threshold ($\geq$27.5) steady-state values found in our study are similar to those of Adib et al. [22], who reported sensitivity, specificity, positive and negative predictive values for IL-6 with a threshold value of 30 pg/mL 78%, 95%, 100% and 87%, respectively, for the diagnosis of neonatal sepsis. For PCT and IL-6, patients in Group 1 have a significantly higher mean compared to the other two groups, whose means are not statistically different from each other. The obtained results demonstrate comparable sensitivity and negative predictive value, albeit at the cost of reduced specificity and negative predictive value. Among the various novel markers investigated in this study, interleukin-6 (IL-6) exhibited the most favorable sensitivity (78%) and negative predictive value (84%).

In the early stage of neonatal bacterial infection, IL-8 levels are also increased. In a study by Boskabadi et al., they showed that serum IL-8 concentration in infants with confirmed sepsis was significantly higher than in healthy infants before blood culture positivity. Also, the serum level of this marker in deceased infants with sepsis was much higher than that of surviving infants. The sensitivity, specificity, positive predictive value and negative predictive value for IL-8 were 95%, 10%, 97% and 10%, respectively, and for CRP was 83%, 86%, 83% and 69%, respectively. The cutoff value of IL8 was above 60 pg/mL [23,24].

In the group of symptomatic infected infants, the mean IL-8 value was 141.98 pg/mL. The serum concentrations of the marker, measured by the described methodology in our study varied within a very wide range, both in the demonstrably ill and in the other two groups of uninfected children. The comparative analysis between the three main groups on the IL-8 value proved that the difference was statistically insignificant.

We also found unsatisfactory results for endocan. This is a new biomarker, being studied in terms of late neonatal sepsis. The mean value in symptomatic infected patients was 163.52 ng/mL. The level in all patients was relatively low and without large variations, compared to results from other authors. The difference between the three groups was not statistically significant and no significant threshold value could be found. It could not be interpreted in terms of diagnostic marker validation criteria.

Our results for endocan are not consistent with those found in the literature, but reports to date are scarce. It is a proteoglycan, secreted by endothelial cells and is suggested to play a role in the pathogenesis of sepsis. An increase in its expression leads to endothelial activation and neovascularization, which are prominent pathophysiological changes associated with inflammation [25].

In a study by Buyuktiryaki et al., CRP, IL-6 and endocan levels were measured in a total of 102 preterm infants. Overall, while all three biomarkers showed "good performance" in differentiating between sepsis and healthy controls, area under the curve (AUC) values in the groups with proven sepsis showed a more significant value for endocan. Further, serial measurements showed that there was no significant difference in CRP and IL-6 levels between the proven and presumed sepsis groups, while endocan levels were significantly higher [26]. Overall, the endocan was found to show a specificity of 94% and a sensitivity of 94.2% [27].

So far, the best combination of markers for predicting late-onset neonatal sepsis with the highest level of validation criteria has not been found. Well known to many clinicians is the neonatal calculator for early onset sepsis (Kaiser Neonatal Sepsis Calculator), combining maternal risk factors and infant's clinical presentation [28]. Similar diagnostic tool for LOS has not yet been approved. Some proposals have been made, such as a Sepsis Prediction Score by Sofouli et al. This is a retrospective study in Greece on 120 newborns with suspected sepsis, combining eight clinical and laboratory parameters such as temperature

instability, feeding volume decrease, platelet count < 150,000/mm$^3$, changes in blood glucose, CRP > 1 mg/dL, circulatory and respiratory deterioration. The scoring model was then validated prospectively on 145 infants [29]. The results were again inconclusive as a lot of septic infants reached very low scores.

We conclude that some interesting findings from our analysis can contribute to the development of a more precise LOS scoring system. The combination of PCT + IL-6 + I:T index demonstrated the highest precision (78%); the result for platelets and the four parameter combination PCT + IL-6 + I:T + PLT was the same. We achieved the best sensitivity (94%) and negative predictive value (95%) in the binary logistic model with two and four variables. Although less precise and specific, the combination of two variables PCT + IL-6 has the same ability to detect individuals with disease and the probability of absence of disease in patients with a negative test (specificity 56%, accuracy 69%) as the more complex combination of variables. This makes it sufficiently valid, especially given its easier applicability. These conclusions can be useful in future research on a bigger cohort.

Last but not least, the study has some limitations. We selected only fpur markers due to the high cost of the kits for determining concentrations via the ELISA assay. The research was institutionally financed and we had set funding limits. For the same reason, we were not able to follow-up the markers in a dynamic manner, rather than just once at the initial phase of infection. We plan, if possible in a future project, to make repeated measurements of the markers to study concentrations over the course of treatment and depending on the severity of the disease, and if sufficient information on these is obtained, we may intersperse our search with other markers. We would also explore cheaper, quicker and more practical methodologies such as the immunochromatographic method. The number of patients is sufficient, but may increase if the project timeframe is extended beyond this (one year).

## 5. Conclusions

The introduction into the routine practice of indicators such as PCT and IL-6 may provide an opportunity for timely diagnosis, the optimization of the therapeutic approach and the reduction in complications from nosocomial infections. The I/T index and platelet count can be relied upon, especially when interpreted together with PCT and IL-6. According to our results, ESM-1 and IL-8 are not reliable markers of late neonatal sepsis. More studies are needed on a larger cohort of newborns as well as on other early biomarkers of inflammation.

**Author Contributions:** Conceptualization, P.G. and V.A.; methodology, A.B.; software, Z.Y.; validation, P.G. and V.A.; formal analysis, Z.Y.; investigation, P.G.; resources, P.G.; data curation, A.B., Z.Y.; writing—original draft preparation, P.G.; writing—review and editing, V.A.; visualization, A.B.; supervision, V.A.; project administration, V.A.; funding acquisition, P.G. All authors have read and agreed to the published version of the manuscript.

**Funding:** This research was funded by Medical university Pleven, grant number D3/2022.

**Institutional Review Board Statement:** The study was conducted in accordance with the Declaration of Helsinki, and approved by the Ethics Committee of Scientific Research of Medical University Pleven (phone №+35964884197; address: Bulgaria, Pleven, Kliment Ohridski Str. #1). Approval Code: 708-CENID/01.06.2022. Approval date: 1 June 2022.

**Informed Consent Statement:** Informed consent was obtained from all subjects involved in the study.

**Data Availability Statement:** Data sharing not applicable. No new data were created or analyzed in this study. Data sharing is not applicable to this article.

**Conflicts of Interest:** The authors declare no conflict of interest.

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
