# Peer review of "Early Diagnostic Markers of Late-Onset Neonatal Sepsis"

_pediatrrep, doi:10.3390/pediatric15030050_

Round 1

Reviewer 1 Report

The manuscript titled: EARLY DIAGNOSTIC MARKERS OF LATE-ONSET SEPSIS OF THE NEWBORN by Preslav Gatsev and others is a really good piece of paper. 

My corrections are more color than substantive.

1. I would edit the title slightly. Especially its second part.

2. why these dates in the abstract? ((January, 13 2022 – January, 2023))

3. Why these numbers in the abstract? (Sixty newborns with an aver-16 age gestational age of 29.75 ± 3.61 gestational weeks were included, of which 35% were symptomatic 17 and infected, 33.3% were ......). After all, the authors refer to them in detail later on anyway.

4. The list of keywords is quite poor, and the work is probably much more interesting. It is worth completing the list of keywords.

5. The introduction is outrageously short. For such an important topic, it should be at least 3 times longer. Even after entering the keywords proposed by the authors, there is a lot to choose from. This has to be supplemented.

6. In addition, the introduction lacks a clearly separated purpose of the work, in the sense of a paragraph with a goal.

Please correct it. From what I see in other authors' works, they know how to write decent introductions.

7. Fig 2 - please correct the axis descriptions - they must be larger. Can not see anything.

8. Summary section - needs to be corrected, what is now considered a joke. Please correct it.

To sum up, the work is really good, only sometimes underdeveloped. After some minor corrections, you can publish it with pleasure.

Author Response

Response to Reviewer 1 Comments

Manuscript ID: pediatrrep-2588146  

EARLY DIAGNOSTIC MARKERS OF LATE-ONSET SEPSIS OF THE NEWBORN

Dear Reviewer,

Thank you for the careful evaluation of our manuscript entitled “EARLY DIAGNOSTIC MARKERS OF LATE-ONSET SEPSIS OF THE NEWBORN”. We have revised the manuscript taking into account the suggested modifications. All changes in the MS are highlighted in yellow.

Point 1: I would edit the title slightly. Especially its second part.

Response 1: Corrected. “…late on-set sepsis of the newborn” was changed to “late-onset neonatal sepsis”

Point 2: Why these dates in the abstract?

Response 2: Corrected.

Point 3: Why these numbers in the abstract?

Response 3: Corrected.

Point 4: The list of keywords is quite poor, and the work is probably much more interesting. It is worth completing the list of keywords.

Response 4: Corrected. I have changed them and added more.

Point 5: The introduction is outrageously short.

Response 5: Corrected. More information has been added with 5 new references.

Point 6: In addition, the introduction lacks a clearly separated purpose of the work, in the sense of a paragraph with a goal.

Response 6: Corrected. The aim of the study was previously in the beginning of section materials and methods, it is now moved to the end of the introduction.

Point 7: Fig 2 - please correct the axis descriptions - they must be larger.

Response 7: Corrected.

Point 8: Summary section - needs to be corrected, what is now considered a joke.

Response 8: Corrected.

Reviewer 2 Report

I am happy to be able to assist with the review of this valuable manuscript.

The reviewer has the following concerns:

#1 The combination of IL-6 and PCT has the potential to be a useful biomarker: In Figure 2, the authors have constructed four ROC curves. The AUC for I:T and PLT is clearly lower than the AUC for IL-6 and PCT. This suggests that if data combining IL-6 and PCT were available, it is highly likely to function sufficiently as a biomarker for sepsis. In other words, the combination of I:T and PLT might not be necessary. The authors are advised to address this counterargument and provide additional insights in the Discussion section.

#2 In the authors' study, two biomarkers, I/T index and PLT, demonstrated high accuracy: Please discuss the reasons for obtaining such results in the Discussion.

N/A

Author Response

Response to Reviewer 2 Comments

Manuscript ID: pediatrrep-2588146  

EARLY DIAGNOSTIC MARKERS OF LATE-ONSET SEPSIS OF THE NEWBORN

Dear Reviewer,

Thank you for the careful evaluation of our manuscript entitled “EARLY DIAGNOSTIC MARKERS OF LATE-ONSET SEPSIS OF THE NEWBORN”. We have revised the manuscript taking into account the suggested modifications. All changes in the MS are highlighted in yellow.

Point 1: The combination of IL-6 and PCT has the potential to be a useful biomarker: In Figure 2, the authors have constructed four ROC curves. The AUC for I:T and PLT is clearly lower than the AUC for IL-6 and PCT. This suggests that if data combining IL-6 and PCT were available, it is highly likely to function sufficiently as a biomarker for sepsis. In other words, the combination of I:T and PLT might not be necessary. The authors are advised to address this counterargument and provide additional insights in the Discussion section.

Response 1: Corrected. We made the new regression model, ROC curve, validation criteria values (Tabl. 4) and corresponding interpretation in the discussion. The proposed combination has lower values of some of the criteria, but it is good enough and more practical.

Point 2: In the authors' study, two biomarkers, I/T index and PLT, demonstrated high accuracy: Please discuss the reasons for obtaining such results in the Discussion.

Response 2: Corrected. It is discussed in the text, we added one more reference concerning the PLT.

Reviewer 3 Report

The authors conducted an interesting study on the validity of a new diagnostic approach to LOS in newborns. This study integrates previous experiences with the use of new biomarkers.

Major comment

1) The Authors report that in Bulgaria only a few Centers routinely test PCT levels (line 229 and following). In fact, this test is the standard test, along with PCR, for diagnosing sepsis. Is this still the current situation in Bulgaria? Please comment on

2) What do you The Authors think about the determination of plasma levels of CRP or PCT, serially in the first days after birth, as a means of diagnosing LOS? Do they have any clinical experience on that modality? Please comment on.

3) Please in the Discussion consider some limitations of the work, like the economic aspects of the choice proposed by them of a panel of 3-4 markers, if they consider it feasible or not too expensive

Good

Author Response

Response to Reviewer 3 Comments

Manuscript ID: pediatrrep-2588146  

EARLY DIAGNOSTIC MARKERS OF LATE-ONSET SEPSIS OF THE NEWBORN

Dear Reviewer,

Thank you for the careful evaluation of our manuscript entitled “EARLY DIAGNOSTIC MARKERS OF LATE-ONSET SEPSIS OF THE NEWBORN”. We have revised the manuscript taking into account the suggested modifications. All changes in the MS are highlighted in yellow.

Point 1: The Authors report that in Bulgaria only a few Centers routinely test PCT levels (line 229 and following). In fact, this test is the standard test, along with PCR, for diagnosing sepsis. Is this still the current situation in Bulgaria? Please comment on

Response 1: Corrected. Yes, the statement from the article is relevant to the situation in the country for the beginning of this year (the time of writing the manuscript). After the end of this study we managed to implement regular and serial measurements of PCT in our hospital but this is a subject to future research. PCR technique is used only for viral causative agents of infections of the nervous system (the LP probe is sent to the capital city of Sofia), so our experience with this is scarce. I commented this point as limitations of the study at the end of the discussion.

Point 2: What do you The Authors think about the determination of plasma levels of CRP or PCT, serially in the first days after birth, as a means of diagnosing LOS? Do they have any clinical experience on that modality? Please comment on

Response 2: Corrected. We still don’t have statistics on serial measurements of PCT since we started it recently. It is now considered a standard test together with serial measurement of CRP but this data can be included in a future study focusing on serial measurements. We could not afford more measurements of the new markers (IL-6,-8, endocan) due to financial reasons and we decided to focus only on the initial phase of infection (also commented as limitation of the study).

Point 3: Please in the Discussion consider some limitations of the work, like the economic aspects of the choice proposed by them of a panel of 3-4 markers, if they consider it feasible or not too expensive

Response 3: Corrected. We added a paragraph on limitations of the study at the end of the discussion.

Round 2

Reviewer 1 Report

The authors improved this manuscript very well, and I recommend this work for publication. 

Reviewer 2 Report

Thank you for involving me in the review of the intriguing revised manuscript.

The authors have addressed my comments and made revisions to the table. The authors have identified a combination of biomarkers that offer higher accuracy compared to the combination of IL6 and PCT, which adds value to publication.

I believe that these revisions have made the manuscript more interesting to a broader readership.

N/A

Reviewer 3 Report

I have no further comments

Good